# Chemical Control of the Invasive Weed *Trianthema portulacastrum*: Nethouse Studies

**DOI:** 10.3390/plants14010019

**Published:** 2024-12-25

**Authors:** Yaakov Goldwasser, Onn Rabinowitz, Jackline Abu-Nasser, Evgeny Smirnov, Guy Achdary, Hanan Eizenberg

**Affiliations:** 1Valley Farmers Center Ltd., Migdal Haemek 2310201, Israel; 2Northern Research and Development, Kiryat Shmona 1101600, Israel; onnrab@gmail.com; 3Newe Ya’ar Research Center, Ramat Yishay 3009500, Israel; jackline@volcani.agri.gov.il (J.A.-N.); evegenysm30@gmail.com (E.S.); achdarig@volcani.agri.gov.il (G.A.); eizenber@volcani.agri.gov.il (H.E.)

**Keywords:** controlled environment studies, desert horse purslane, herbicides, Hula Valley Israel, invasive plants, pre-emergence, post-emergence

## Abstract

*Trianthema portulacastrum* L. (Aizoaceae), commonly known as desert horse purslane or black pigweed, is a C4 dicot succulent invasive annual plant that is widespread in agricultural fields in Southeast Asia, tropical America, Africa, and Australia. In Israel, *Trianthema portulacastrum* is an invasive weed of increasing importance in agricultural fields, including mainly corn, tomato, alfalfa watermelon, and groundnut crops. The significance of this weed in crops has been recently reported in neighboring countries of Jordan and Egypt. In previous studies, we have examined and described the spread, biology, and germination requirements of *Trianthema portulacastrum* in Israel. The present study aimed to investigate the efficiency of single pre- and post-emergence herbicides and the combination of pre-applied herbicides for the control of this invasive weed in pots in a nethouse. We conducted three sequential experiments in a nethouse: (1) screening of pre-emergence herbicides, (2) screening of post-emergence herbicides, and (3) assessment of residual activity of combined pre-emergence herbicides in three distinct Hula Valley soil types. Efficacy was evaluated through weekly assessments of seedling emergence and vigor, with the final shoot fresh weight determined upon the experiment’s completion. In all experiments, weekly counts and vigor estimation of *T. portulacastrum* seedlings were conducted, and shoot fresh weights were determined at the end of the experiments. The results of pre-emergence herbicide screening showed that Fomesafen, Terbutryne, Flurochloridon, Sulfosulfuron, Cyrosulfamid + Izoxaflutole, and Dimethenamid were the most effective herbicides, leading to complete eradication of *T. portulacastrum* plants. Results of the post-emergence screening revealed that Saflufenacil, Foramsulfuron, Tembotrione + Isoxdifen-ethyl, and Rimsulfurom Methyl completely controlled the weed. In the soil residual study, three herbicide combinations (Fomesafen + Terbutryn, Sulfosulfuron + Fomesafen, and Dimethenamid + Flurochloridon) provided effective control across all soil types. These findings provide a foundation for future field trials investigating integrated pre- and post-emergence herbicide programs for *T. portulacastrum* management in various crops.

## 1. Introduction

*Trianthema portulacastrum* L. (Aizoaceae), commonly known as desert horse purslane or black pigweed, is a widespread weed in Southeast Asia, America, Africa, Australia, and Asia, with no known center of origin [1] (Figure 1). It is a branched annual C4 dicot plant with ovate, simple leaves, and pink, five-petal bisexual flowers that are insect-pollinated. The flowers produce capsules that release many small, kidney-shaped, hard-coated seeds throughout the warm spring–autumn period. Seed dispersal occurs through multiple vectors, including soil movement, water flow, and agricultural equipment, with seeds maintaining viability in the soil for several years [1,2,3]. The agricultural impact of *T. portulacastrum* spans diverse cropping systems worldwide, with recent research documenting significant infestations in groundnuts [4], rice [5,6,7,8,9,10,11,12], pearl millet [13], and greengram [14,15,16]. *T. portulacastrum* has documented medicinal properties, including antimicrobial, analgesic, anti-inflammatory, anti-hyperglycemic, and hepatoprotective activities [17,18,19,20].

There have been numerous reports of this weed as an invasive species in the Middle East and the Indian subcontinent. In Egypt, *T. portulacastrum* was reported to be an invasive weed in field crops and orchards, with high phenotypic differences resulting from variable soil moisture, soil salinity, and temperature regimes in diverse geographical regions [21]. Controlled-environment seed germination studies have revealed impressive reproductive capacity, with individual plants producing an average of 1931 seeds, each weighing an average of 1.08 g in dry weight. A different study revealed that high temperatures of 30–45 °C stimulate maximum seed germination [22]. Reports from India and Pakistan have also described this plant as an invasive and noxious weed in many crops, with few effective control measures. A study of chemical and cultural control measures indicated that efficient weed control was achieved when these measures were applied in the early *T. portulacastrum* growth stages, up to 40 days after emergence [23]. Attempts at cultural and biological measures to control this troublesome weed have reported only limited success [24,25,26].

Recent reports on the invasion of this weed into Jordan and Israel led to the inclusion of *T. portulacastrum* on the 2019 invasive weed alert list of the European and Mediterranean Plant Protection Organization (EPPO) [27]. In Israel, *T. portulacastrum* field infestation throughout agricultural regions has been reported in recent years. This invasive weed thrives in warm, moist soil environments, and it is acclimated to the agricultural conditions of irrigated crops in Israel, especially in the hot and moist conditions characteristic of the Hula Valley region.

In previous studies, we surveyed and mapped *T. portulacastrum* populations in the Hula Valley in Israel, examined the biology of the plant, collected seeds for laboratory and greenhouse studies, determined seedling emergence depths, and studied the effects of temperature and light on germination, leading to the development of a temperature-based seed germination model [3].

The aim of the present study was to screen in pots under nethouse conditions single pre- and post-emergence herbicides and pre-emergence herbicide combinations in different soils of the Hula Valley in Israel for sustainable control of *T. portulacastrum*. This research provides a foundation for developing practical field-scale management strategies for this increasingly important invasive weed in Israel and in other global infested regions.

## 2. Materials and Methods

### 2.1. Screening Herbicides for T. portulacastrum Control

Screening for effective herbicides for the control of *T. portulacastrum* was conducted in 0.25 L pots containing Newe Ya’ar heavy soil placed in a nethouse of 50 mesh at the Newe Ya’ar Research Center. *T. portulacastrum* seeds used in this study were collected in the Hula Valley in 2019–2020, immersed at the beginning of each experiment in 97% sulfuric acid for 30 min, and then thoroughly washed with sterile water to ensure the maximum germination rate.

Herbicide applications were conducted with a “Resses” table sprayer delivering 200 L/ha. Following application, pots were placed in a nethouse and irrigated daily using mini sprinklers.

### 2.2. Pre-Emergence Herbicide Screening

For the pre-emergence experiment, *T. portulacastrum* seeds were treated for 30 min with 97% sulfuric acid and sown in pots at 2 cm soil depth in Newe Ya’ar soil. Seeding was performed with 10 seeds per pot with five replication pots per treatment. Herbicides were then applied to the soil in the pots, followed by the placement of pots in a nethouse (Figure 2). In the treatment in which Dimethenamid was applied, the herbicide was sprayed on the soil surface, followed by its incorporation throughout the soil. *T. portulacastrum* plant vitality assessments were conducted weekly, with final shoot fresh weights determined 32 days after application (DAA) by harvesting plants at the soil level.

### 2.3. Post-Emergence Herbicide Screening

For the post-emergence experiment, *T. portulacastrum* seeds were sown as described in Experiment 1, but with six replicate pots per treatment. Herbicide applications were made when plants reached the four-true-leaf stage (13 days after sowing). Weekly assessments documented plant vigor and survival rates, with final shoot fresh weights determined 32 DAA. See Figure 3.

### 2.4. Pre-Emergence Herbicide Residual Effects on Three Hula Valley Soils

This experiment evaluated herbicide combination treatments across three representative Hula Valley soil types Menara, Shamir, and Lehavot soils (see Table 1) in 800 cm^3^ pots with four replicates per treatment. Seeds were sown at 2 cm depth at three intervals (0, 25, and 50 DAA) to assess treatment longevity and placed at the Newe-Ya’ar nethouse. Weekly monitoring of plant vigor and survival continued until 50 days after treatment (DAT) for each seeding date, concluding with fresh weight determination. Pre-emergence application of herbicides at each application date was performed using the Newe Ya’Newear table sprayer delivering 200 L/ha, followed by the return of the pots to the nethouse. The pots were irrigated daily using a mini-sprinkler system. Assessment of *T. portulacastrum* plant vigor and the number in each pot was conducted weekly, and, at the end of the experiment, 50 DAT for each seeding date, plants from each pot were cut at soil height and their fresh weight determined.

## 3. Results

### 3.1. Pre-Emergence Herbicide Screening Assay

The weekly estimations of *T. portulacastrum* plant vigor conducted for the pre-emergence pot trial reveal that at the end of the experiment, 32 DAA, Fomesafen, Terbutryne, Flurochloridon, Sulfosulfuron, Cyprosulfamid + Izoxaflutole, and Dimethenamid controlled 95–100% of the weed infestation throughout the experiment, Saflufenacil application resulted in partial control, and Imazamox was the least effective (Figure 4). At the end of the experiment, 32 DAA, all herbicide treatments excluding Imazamox drastically reduced the *T. portulacastrum* final shoot fresh weight in comparison to the non-treated control, whereas Fomesafen and Cyprosulfamid + Izoxaflutole applications led to 100% eradication of the plants (Figure 5).

**Figure 4 plants-14-00019-f004:**
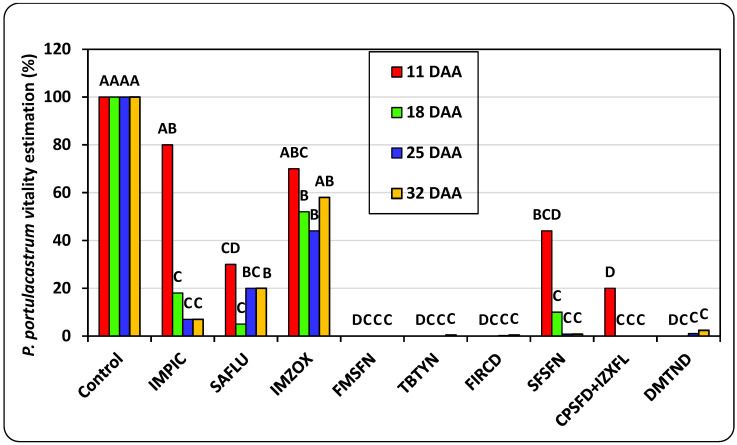
Weekly vitality estimations of *T. portulacastrum* plants in the pre-emergence pot experiment. Bars at each DAA topped with different letters are statistically different according to Tukey–Kramer HSD, *p* = 0.01. See Table 2 for full herbicide chemical names and application rates. DAA: days after application.

**Table 2 plants-14-00019-t002:** Herbicides tested in the pre- and post-emergence nethouse screening studies.

Herbicide Chemical Name	Abbreviation	Herbicide Trade Name ^a^	Active Ingredient	Application Rate g/ha
Aclonifen	ACNFN	Challenge	600 g/L	1200
Bentazon	BNTON	Bazagran	480 g/L	1920
Cyprosulfamid+ Izoxaflutole	CPSFD+IZXFL	Balance Smart	240 g/L+ 240 g/L	72+ 72
Dimethenamid	DMTND	Frontier Optima	720 g/L	2520
Flurochloridon	FIRCD	Racer	250 g/L	625
Fomesafen	FMSFN	Flex, Relax	250 g/L	625
Foramsulfuron	FRSFN	Equip	22.5 g/L	45
Imazamox	IMZOX	Pulsar	40 g/L	32
Imazapic	IMPIC	Cadre	240 g/L	480
Metribuzin	MTBZN	Sencor	70%	21
S-Metolachlor	MTLCR	Dual S Gold	915 g/L	119
Pyraflufen	PYFFN	Ecopart	20 g/L	10
Rimsulfuron Methyl	RMSFM	Titus	25%	37.5
Saflufenacil	SAFLU	Heat	70%	35
Sulfosulfuron	SFSFN	Monitor	75%	38
Tembotrione+ Isoxdifen-ethyl	TMBON+ ISXFE	Laudis	44 g/L+ 22 g/L	110+ 55
Terbutryne	TBTYN	Terbutrex	500 g/L	1000

^a^ Herbicide trade names, as registered in Israel.

### 3.2. Post-Emergence Herbicide Assay

The weekly assessments of plant vigor conducted for the post-emergence pot trial revealed that the herbicides Saflufenacil, Foramsulfuron, Tembotrione + Isoxdifen-ethyl, Pyraflufen, and Rimsulfuron Methyl were the most effective in *T. portulacastrum* control throughout the trial, while Imazamox and Aclonifen were not effective (Figure 6). At the end of the experiment, 32 DAA, all herbicide treatments statistically reduced the *T. portulacastrum* final shoot fresh weight in comparison to the non-treated control, whereas Saflufenacil, Foramsulfuron, and Tembotrione + Isoxdifen-ethyl were the most effective, leading to 100% eradication of the plants (Figure 7).

### 3.3. Pre-Emergence Herbicide Residual Effect on Three Hula Valley Soils

The final shoot fresh weight of *T. portulacastrum* shoots in the three Hula Valley soils 0 DAA of pre-emergence applied herbicides is exhibited in Figure 8, Figure 9 and Figure 10, while the final shoot fresh weight in the three soils 50 DAA is shown in Figure 11, Figure 12 and Figure 13. In the Lehavot soil, at 0 DAA, all herbicide combinations effectively and significantly provided complete control of the weed, whereas the non-treated control yielded an average of 14.3 g F.W of *T. portulacastrum* per pot (Figure 8). In the Menara soil, at 0 DAA, all herbicide combinations effectively and significantly controlled the weed, excluding the Saflufenacil + Terbutryne treatment, which yielded 25.8 g per pot F.W. of *T. portulacastrum* compared to 38.4 g in the control treatment (Figure 9).

In the Shamir soil, all herbicide combinations effectively and significantly controlled the weed at 0 DAA, excluding the Rimsulfuron Methyl + Metribuzin treatment, which yielded 11.8 g per pot F.W. of *T. portulacastrum* compared to 60.4 g per pot F.W. of *T. portulacastrum* in the control treatment. (Figure 10).

Overall, these results demonstrate that most pre-emergence herbicides tested resulted in excellent control of *T. portulacastrum* at 0 DAA, excluding Saflufenacil + Terbutryn in the Menara soil and rimsulfuron Methyl + Metribuzin in the Shamir soil.

The results of *T. portulacastrum* control by the herbicide combinations at 50 DAA indicate a decrease in control efficacy. However, the combinations of Sulfosulfuron + Fomesafen and Fomesafen + Terbutryne effectively and significantly controlled the weed in the Lehavot and the Shamir soils (Figure 11 and Figure 13), while in the Menara soil all herbicide combinations significantly controlled the weed (Figure 12).

A table summarizing the different herbicides used in this study, including their chemical names, trade names, and registered crops in Israel, is presented in Table 3. The most effective herbicides identified in this study are outlined in blue, while the Israeli-registered crops for each herbicide are outlined in yellow. This table shows that effective and selective herbicides are available for *T. portulacastrum* control across various crops. Specifically for the crops in the Hula Valley, the selective and efficient registered herbicides are for alfalfa, tomato, maize, and groundnuts.

## 4. Discussion

Based on our research on the biology and germination traits of the rapidly spreading invasive weed *T. portulacastrum* in Israel [3], the present study aimed to conduct nethouse studies for the screening of effective pre- and post-emergence herbicides aimed at controlling this troublesome weed in crops. The effective pre-emergence herbicides that were tested in this study for controlling *T. portulacastrum* were Fomesafen, Terbutryne, Flurochloridon, Sulfosulfuron, Cyrosulfamid + Izoxaflutole, and Dimethenamid. The most effective post-emergence herbicides were Saflufenacil, Foramsulfuron, Tembotrione + Isoxdifen-ethyl, and Rimsulfuron Methyl. In the final nethouse experiment using three representative Hula Valley soils, three pre-emergence herbicide combinations demonstrated superior control of *Trianthema portulacastrum* across all soil types: Fomesafen + Terbutryne, Sulfosulfuron + Fomesafen, and Dimethenamid + Flurochloridon. Of these, the first two combinations are approved for weed management in processing tomatoes, while the third is authorized for use in groundnut cultivation.

Chemical control trials of *T. portulacastrum* have been reported in various crops by multiple researchers, but only two of these crops are grown in the Hula Valley and Israel: maize and groundnuts. In groundnuts, a recent comprehensive review summarizes the literature on the overall herbicide efficacy against *T. portulacastrum* infestations in groundnut experiments over the last 20 years [4]. However, the available literature demonstrates a substantial range in herbicide efficacy due to differences in environmental conditions and fluctuations in weed germination and infestation across years and locations. Grichar [2,28] reports experiments in groundnuts in the US and provides the following results for pre-emergence regimes: Dimethenamid (51%), Pendimethalin (67%), and Flumioxazin (64–73%). For post-emergence regimes, the efficacy is as follows: Acifluorfen (17–77%), Lactofen (68–100%), 2,4-DB (13–84%), Imazethapyr (65%), Diclosulam (7–73%), Imazapic (7–62%), and Imazethapyr (13–27%). While several herbicides are available for weed control in groundnut, no single herbicide can provide season-long weed control due to limited application timing, lack of extended residual activity, variability in the weed control spectrum, and rotational restrictions. Consequently, effective weed management in groundnuts necessitates the use of herbicide mixtures and/or sequential application of pre-plant-incorporated, pre-emergence, and/or post-emergence herbicides, as well as residual herbicides. Groundnut has a long growing season (140 to 160 days) for development and maturity [29], and residual herbicides allow weeds to be more competitive for light, particularly during the early stage of crop growth [30].

## 5. Conclusions

This study provides valuable insights into the chemical control of the invasive weed *Trianthema portulacastrum* through a systematic evaluation of pre-emergence, post-emergence, and residual herbicide treatments under controlled nethouse conditions. Based on global observations across multiple crops, including our findings, effective management of *T. portulacastrum* requires a combination of herbicide treatments due to its continuous germination throughout the growing season and declining herbicide efficacy over time. This management approach necessitates both herbicide mixtures and sequential applications, including pre-plant incorporated (PPI), preemergence, and postemergence. This challenge is particularly pronounced in slow-growing crops with sparse canopy coverage, which create favorable conditions for both continuous *T. portulacastrum* germination and rapid establishment of this aggressive C4 weed. Under nethouse conditions, pot experiments demonstrated effective *T. portulacastrum* control using several herbicides. Preemergence treatments, including Fomesafen, Terbutryne, Flurochloridon, Sulfosulfuron, Cyrosulfamid + Izoxaflutole, and Dimethenamid, provided successful control, as did postemergence applications of Saflufenacil, Foramsulfuron, Tembotrione + Isoxdifen-ethyl, and Rimsulfuron Methyl. Furthermore, herbicide combinations of Fomesafen + Terbutryne and Sulfosulfuron + Fomesafen demonstrated high efficacy at 50 days after application (DAA) when tested across three representative soil types from the Hula Valley. In recent years, *T. portulacastrum* infestations have increased significantly and are spreading to new regions. Thus, following our studies and our understanding of the biological traits of this invasive weed, the major importance of the present research was the investigation of control measures for pre- and post-applied selective herbicides and herbicide combinations that could be employed safely in different crops. By combining our understanding of this invasive weed’s biological characteristics with extensive herbicide trials, we identified effective selective herbicides and herbicide combinations that can be safely implemented across diverse cropping systems. Nevertheless, the risks arising from chemical residues of plant protection products in edible plant tissues, particularly those resulting from complex agrochemical “cocktails”, are limited by chemical regulation standards, thus warranting comprehensive examination.

Our findings establish a foundation for developing comprehensive management strategies, although field-scale validation remains essential. Future research priorities include optimizing application rates and timing, conducting extensive field trials, and integrating herbicide applications with cultural control practices. Based on the findings of the present study, detailed elements of the *T. portulacastrum* control strategy will be studied under field conditions.

## Figures and Tables

**Figure 1 plants-14-00019-f001:**
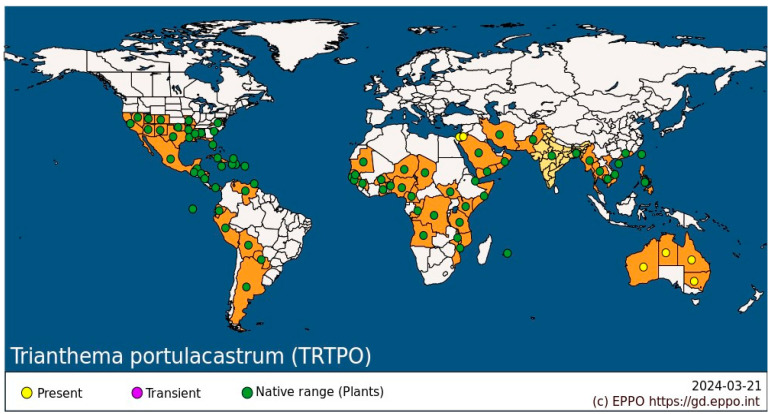
Global distribution of *T. portulacastrum*, updated March 2024, according to the European and Mediterranean Plant Protection Organization (EPPO) Global Database. https://gd.eppo.int.

**Figure 2 plants-14-00019-f002:**
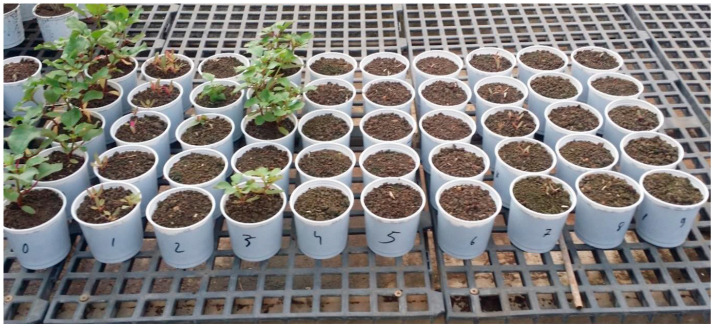
Pots of the pre-emergence trial at 18 DAA in the nethouse, 5 replications = 5 pots per treatment. 0 = control, 1 = Dimethenamid 2520 g/ha, 2 = Imazapic 480 g/ha, 3 = imazamox 32 g/ha, 4 = Saflufenacil 35 g/ha, 5 = Fomesafen 625 g/ha, 6 = Terbutryne 1000 g/h, 7 = Flurochloridon 625 g/ha, 8 = Sulfosulfuron 38 g/ha, 9 = Cyprosulfamid 72 g/ha + Izoxaflutole 72 g/ha.

**Figure 3 plants-14-00019-f003:**
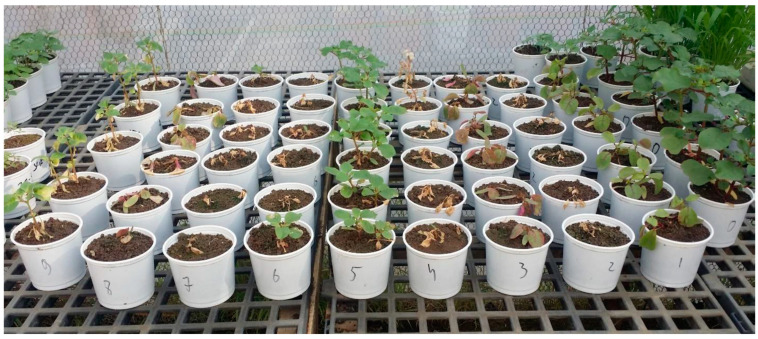
Pots of the post-emergence trial at 8 DAA in the nethouse, 6 replications with 6 pots per treatment. Herbicide and application rate: 0 = control, 1 = Imazapic 480, 2 = Saflufenacil 35 g/ha, 3 = Foramsulfuron 45 g/ha, 4 = Tembotrione 44 g/ha + Isoxdifen-ethyl 110 g/ha, 5 = Imazamox 32 g/ha, 6 = Bentazon 1920 g/ha, 7 = Pyraflufen 10 g/ha, 8 = Rimsulfuron Methyl 37.5 g/ha, 9 = Aclonifen 1200 g/ha.

**Figure 5 plants-14-00019-f005:**
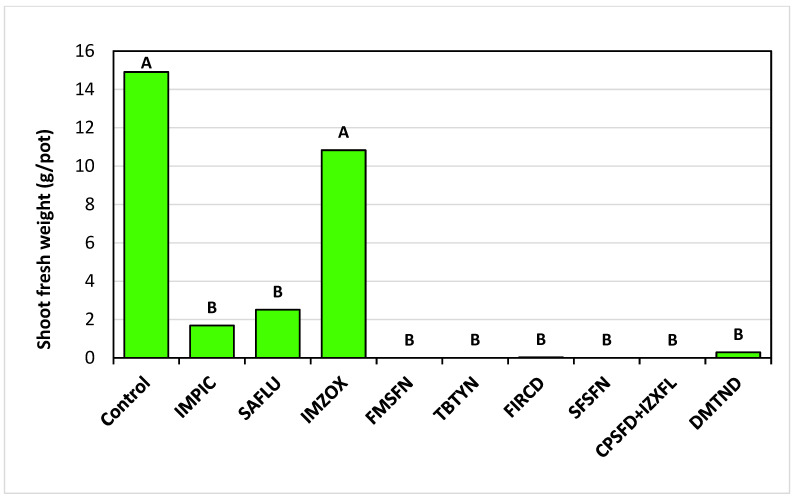
*T. portulacastrum* final shoot fresh weight at the termination of the pre-emergence pot experiment, 32 DAA. See Table 2 for full herbicide chemical names and application rates. Bars topped with different letters are statistically different according to Tukey–Kramer HSD, *p* = 0.01.

**Figure 6 plants-14-00019-f006:**
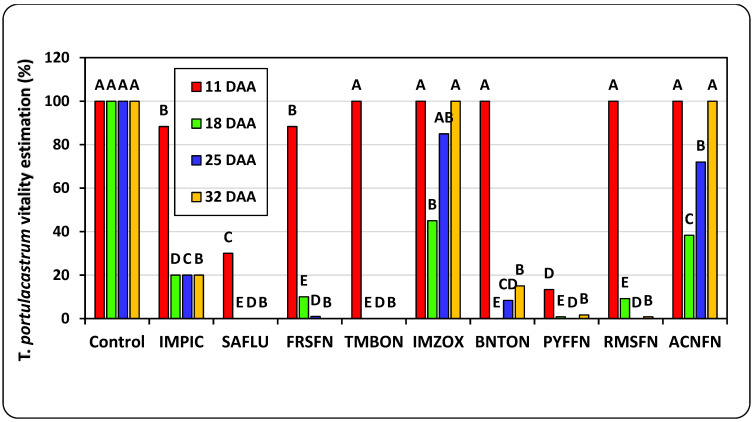
Weekly vitality estimations of *T. portulacastrum* plants in the post-emergence pot experiment. Herbicides were applied on *T. portulacastrum* at the 4-leaf stage on August 23rd, 13 days after seeding. See Table 2 for full herbicide chemical names and application rates. Bars at each DAA topped with different letters are statistically different according to Tukey–Kramer HSD, *p* = 0.01.

**Figure 7 plants-14-00019-f007:**
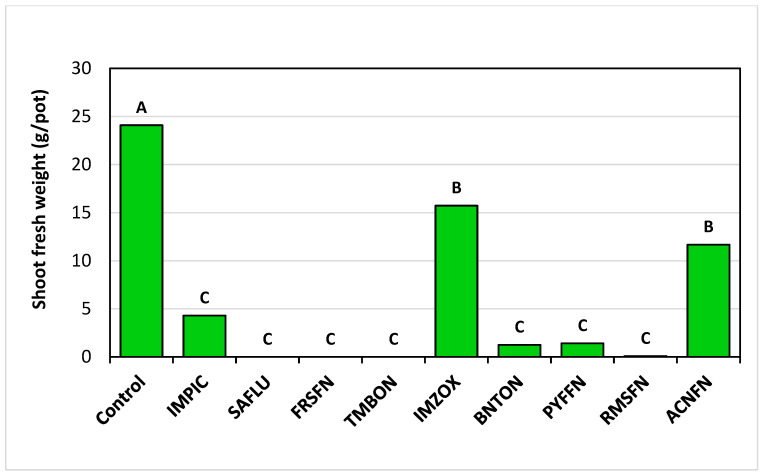
*T. portulacastrum* shoot fresh weights at the termination of the post-emergence applied herbicides experiment, 32 DAA. See Table 2 for full herbicide chemical names and application rates. Bars topped with different letters are statistically different according to Tukey–Kramer HSD, *p* = 0.01.

**Figure 8 plants-14-00019-f008:**
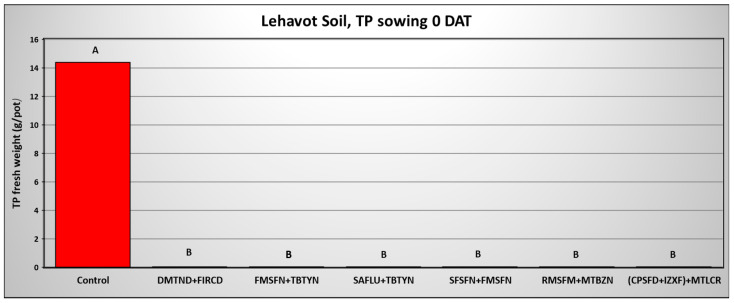
*T. portulacastrum* final shoot fresh weight at the termination of the pre-emergence pot experiment in Lehavot soil seeded 0 days after herbicide application. See Table 2 for herbicide abbreviations and rates. Bars topped with different letters are statistically different according to Tukey–Kramer HD, *p* = 0.01.

**Figure 9 plants-14-00019-f009:**
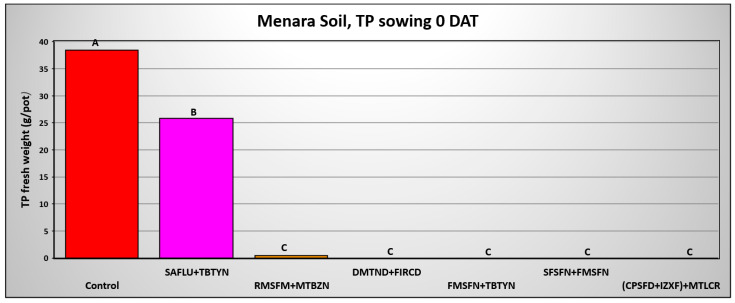
*T. portulacastrum* final shoot fresh weight at the termination of the pre-emergence pot experiment in Menara soil seeded 0 days after herbicide application. See Table 2 for herbicide abbreviations and rates. Bars topped with different letters are statistically different according to Tukey–Kramer HD, *p* = 0.01.

**Figure 10 plants-14-00019-f010:**
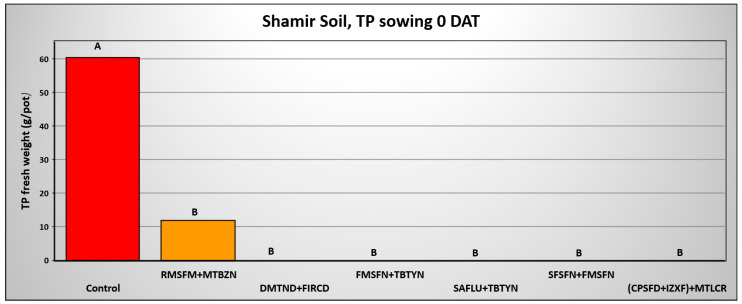
*T. portulacastrum* final shoot fresh weight at the termination of the pre-emergence pot experiment in Shamir soil seeded 0 days after herbicide application. See Table 2 for herbicide abbreviations and rates. Bars topped with different letters are statistically different according to Tukey–Kramer HD, *p* = 0.01.

**Figure 11 plants-14-00019-f011:**
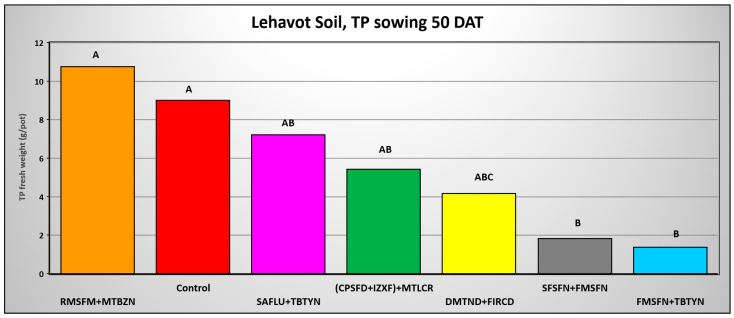
*T. portulacastrum* final shoot fresh weight at the termination of the pre-emergence pot experiment in Lehavot soil seeded 50 days after herbicide application. See Table 2 for herbicide abbreviations and rates. Bars topped with different letters are statistically different according to Tukey–Kramer HD, *p* = 0.01.

**Figure 12 plants-14-00019-f012:**
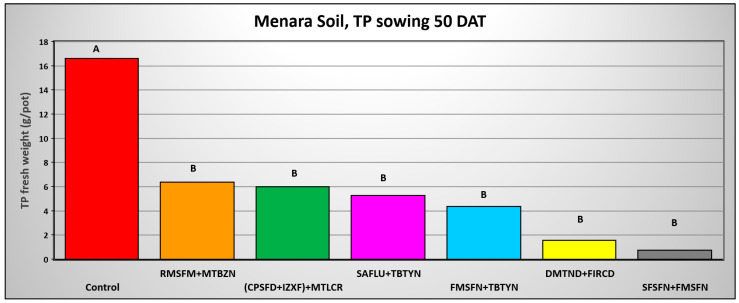
*T. portulacastrum* final shoot fresh weight at the termination of the pre-emergence pot experiment in Menara soil seeded 50 days after herbicide application. See Table 2 for herbicide abbreviations and rates. Bars topped with different letters are statistically different according to Tukey–Kramer HSD, *p* = 0.01.

**Figure 13 plants-14-00019-f013:**
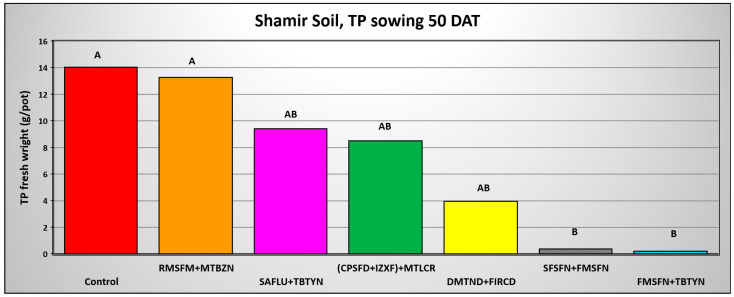
*T. portulacastrum* final shoot fresh weight at the termination of the pre-emergence pot experiment in Shamir soil seeded 50 days after herbicide application. See Table 2 for herbicide abbreviations and rates. Bars topped with different letters are statistically different according to Tukey–Kramer HSD, *p* = 0.01.

**Table 1 plants-14-00019-t001:** Soil properties of the Newe Ya’ar soil and the three Hula Valley soils used in the pre-emergence herbicide residual effect experiment.

Soil Location	Location Coordinates	Total Lime (%)	Active Lime (%)	Sand (%)	Silt (%)	Clay (%)	pH	Organic Matter (%)
Newe Ya’ar	32°42’13.6” N 35°11′05.7” E	-	12.5	15.2	26.3	58.6	7.5	2.0
Lehavot	33°8’13.46” N 35°38’25.14” E	22.1	9.6	11.0	38.4	50.6	7.5	1.94
Shamir	33°4’42.87” N 35°37’15.24” E	67.3	18.4	15.0	46.4	38.6	7.4	3.45
Menara	33°6’20.14” N 35°35’0.5” E	40.0	13.0	23.0	34.4	42.6	7.4	7.68

**Table 3 plants-14-00019-t003:** Herbicide chemical and trade names and their registered crops in Israel.

Herbicide Chemical Name	Herbicide Trade Name	Registered Crops in Israel
Aclonifen	Challenge	Parsley, tomato ^b^, onion, chickpea, sweet potato, dill, coriander
Bentazon	Bazagran	Wheat, pea, groundnuts, onion, green bean, vetch
Cyprosulfamid+ Izoxaflutole ^a^	Balance Smart	Maize, chickpea
Dimethenamid	Frontier Optima	Groundnut’s, potato, maize
Flurochloridon	Racer	Carrot, chickpea, sunflower, parsley, coriander, dill, cotton, sweet potato
Fomesafen	Flex, Relax	Peas, chickpea, green beans,
Foramsulfuron	Equip	Maize
Imazamox	Pulsar	Alfalfa, green beans, peas, clover, vetch, groundnuts
Imazapic	Cadre	Groundnuts, tomato
Metribuzin	Sencor	Carrot, potato, tomato
S- Metolachlor	Dual S Gold	Maize, green beans, sunflower, groundnuts, paprika, chickpea
Pyraflufen	Ecopart	Wheat, vetch, onion, groundnuts, orchards
Rimsulfuron Methyl	Titus	Tomato, maize, potato
Saflufenacil	Heat	Wheat, maize, orchards, maize
Sulfosulfuron	Monitor	Wheat, tomato, lawn
Tembotrione + Isoxdifen-ethyl	Laudis	Maize
Terbutryn	Terbutrex	Groundnuts, chickpea, sweet pea, sunflower

^a^ Outlined in blue are the outstanding herbicides for *T. portulacastrum* control, as observed in the present nethouse study. ^b^ Outlined in yellow are the common crops grown in the Hula Valley for which further field trials will be conducted.

## Data Availability

The original contributions presented in this study are included in the article. Further inquiries can be directed to the corresponding author.

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
