# Peer review of "Chemical Control of the Invasive Weed Trianthema portulacastrum: Nethouse Studies"

_plants, 2024, doi:10.3390/plants14010019_

Round 1

Reviewer 1 Report

Comments and Suggestions for Authors

Review of the manuscript entitled „Chemical Control of the Invasive Weed Trianthema portulacastrum – nethouse Studies”

  In the reviewed manuscript, the authors present findings regarding the usefulness of selected pre- and post-emergence herbicides in the control of the dicotyledonous, invasive weed Trianthema portulacastrum. The choice of the study topic due to the threats associated with mass infestation of this weed in countries with a warm climate is fully justified. References regarding weed biology and control methods introduce the reader well to the issue. The conducted screening research was conducted as three pot experiments under controlled conditions in a nethouse. The methodology is routine for this type of research. The experimental design, equipment used, the method of statistical analysis of results are correct. The method of presentation of results does not raise any objections. The figures placed in the text, in the form of bar graphs are a good illustration of the obtained results. The conclusions are confirmed by the results. The manuscript contains new findings regarding T. portulacastrum control of great practical importance. The strength of the article is its clear and correct English.

In my opinion, the manuscript may be the subject of further editorial work.

Detailed comments

l. 275-277: “In a recent publication Grichar et al, 2024) the following combinations were found to be the most effective in T. portulacastrum control: (Grichar 2024)”. The sentence looks unfinished – which combinations?

The "Conclusions" chapter is too long. It should contain, first of all, the most important findings from the present study. The penultimate paragraph in particular does not contribute much to this part of the article. The scope of future studies should rather be outlined in the Discussion, and in the conclusions it should be summed up in general terms, “Based on the findings of the present study, detailed elements of the T. portulacastrum control strategy will be developed in field conditions”.

The authors write that in order to ensure effective control against T. portulacastrum, intensive use of herbicides is recommended, together with sequential applications and the use of mixtures (l. 280-283). In the Discussion, it is worth mentioning very briefly the risks associated with the occurrence of chemical residues of plant protection products in edible parts of plants and their interactions, after the use of agrochemical “cocktails” and t and point out that residue status should also be investigated.In some countries, the maximum number of active substances, the residues of which are found in agricultural crops, is limited by standards.

Detailed comments are of no substantive importance and do not detract from the very high value of the manuscript.

Date of manuscript received: 29 November 2024

Date of this review: 2 December 2024

Author Response

Please see attached file- Response to Reviewer 1

Reviewer 2 Report

Comments and Suggestions for Authors

This is a practical manuscript on screening herbicides for chemical control of invasive alien plants. 

major concern:

1. As for the chemical control of this alien species, the previous work should be described in detail in the preface. In addition, the differences and scientific problems between this experiment and previous experiments should be pointed out through comparison with previous work.

minor suggestion:

2. In Figure 4, Note should be more detailed, such as the meaning of such as CC symbols.

Author Response

 Please see attached file- Response to Reviewer 2

Reviewer 3 Report

Comments and Suggestions for Authors

General comments

This study tackles the challenge of managing the invasive weed Trianthema portulacastrum, a significant threat to agricultural productivity. By evaluating pre- and post-emergence herbicides and their residual effects, it addresses a critical gap in weed control. The findings emphasize the weed impact on crop yields and offer practical solutions for integrated management, particularly in maize, groundnuts, and alfalfa.

Specific comments

Keywords: The terms “Chemical control” and “Trianthema portulacastrum” are already included in the manuscript title. To improve discoverability and impact within the scientific community, we recommend replacing these with more specific keywords that reflect unique aspects of the study.

Introduction: Line 88: Before the aims of the work. To strengthen this section, it is recommended to clearly and concisely define the central research question at the end. Additionally, emphasizing the novelty of the study before presenting the hypothesis will help highlight its originality. Currently, the work may appear incremental due to the absence of a clearly stated hypothesis. It is advised to address the following questions: What is the novelty of this study? What hypothesis is being tested? In addition, perhaps it would be valuable to include a brief text explicitly linking the research objectives to regional or global agricultural sustainability goals. This addition could provide greater depth and relevance to the study.

Methodology: The methodology is robust, using systematic pot trials with clearly defined controls, replicates, and appropriate statistical analyses. The identification of effective herbicides and their combinations under controlled conditions enhances the practical applicability of the findings.

Results: The manuscript presents detailed results, including weekly vitality assessments, fresh weight measurements, and statistical validations. Visuals such as graphs and tables effectively support the text, aiding in data interpretation. However, Figures 8 to 13 should follow the same formats as Figures 4 to 7, without gray backgrounds to improve visual clarity. In addition, gridlines of all figures could be removed for a cleaner presentation. 

Discussion: The discussion is relevant and aligns well with the results. However, it could be improved by linking the findings more explicitly to broader agricultural practices, identifying study limitations, and suggesting future research directions.

Conclusions
This section should be concise and avoid repetition from earlier parts of the manuscript. Summarizing key insights in a succinct manner will enhance readability. Conclude with a compelling final statement that underscores the contribution of the work or future applications.

Interest to Readers
The manuscript is likely to interest scientists working in weed management, herbicide efficacy, and agricultural productivity. It may also provide practical insights for farmers dealing with invasive weeds.

English level: The text has a clear flow that is appropriate and understandable for me. However, certain sentences are overly complex or lack fluency, and some grammar issues need correction for smoother reading. Before possible publication, the authors and editorial staff should carefully revise the work. 

Author Response

Please see attached file- Response to Reviewer 3

Round 2

Reviewer 3 Report

Comments and Suggestions for Authors

The manuscript may be accepted for publication. However, Figure 7 will need to be adjusted as it is misaligned.